# Effect of Pore Characteristics in Polyvinylidene Fluoride/Fumed Silica Membranes on Mass Flux in Solar-Assisted Evaporation Applications

**Mona Bahman, Maryam AlNahyan, Ibrahim Mustafa and Faisal AlMarzooqi \***

Center for Membranes & Advanced Water Technology, Department of Chemical Engineering, Masdar Institute, Khalifa University, P.O. Box 54224, Masdar City, Abu Dhabi, UAE

\* Correspondence: faisal.almarzooqi@ku.ac.ae; Tel.: +971-2-810-9128; Fax: +971-2-810-9901

**Featured Application: Solar Evaporation.**

**Abstract:** Although important, very little has been demonstrated in the literature to experimentally demonstrate the effects of porosities and pore size on the evaporation flux in polymeric membranes. Additionally, we suspect that a batch-mode setup, i.e., stagnant water, could cause a build-up of heat in the system, influencing the evaporation mass-flux mechanism, and jeopardizing the ability to attain a real correlation between evaporation and effects of pore characteristics. Herein, we fabricate polyvinylidene fluoride membranes containing variable amounts of a Fumed Silica additive to achieve membranes with variable properties, and we investigate the change in the performance of the solar-assisted thin-film evaporation utilizing an in-house built continuous flow evaporation setup (to avoid heat build-up effects in the bulk of the water and demonstrate a continuous flow system). Our membrane design approach had two important advantages: (1) the achievement of similar heat transfer and solar absorbance properties and (2) the achievement of variable pore sizes and volume porosities. We show that the mass flux increased as the mean pore size decreased, indicating that the mode of mass transfer occurred due to the thin-film region of the meniscus from the small fluid velocities near the interface, and we attribute the results to the increase in the capillary pumping effects through the mesoporous channels as they get thinner.

**Keywords:** nano-transport; evaporation; solar; desalination; membrane

---

## 1. Introduction

The International Energy Association (IEA) reported that more than 151.19 TWh of energy was consumed in 2016 for desalination in the United Arab Emirates [1]. This great demand reflects the water-energy nexus being faced by several countries throughout the world and stresses the need for research and the development of low-energy consuming desalination technologies. Desalination technologies are typically based on either thermal or membrane-based mechanisms, both of which have been reviewed extensively in the literature [2–7]. There has been a growing interest in thin-film evaporation technologies for cooling applications, electronic devices, perspiration, and more recently, for solar-assisted water desalination because they enable cost-efficient water vapor generation by utilizing solar irradiation in a relatively simple mechanism [8–10].

Thin-film evaporation utilized in solar-assisted desalination is governed by a phase change process in which water first penetrates into the nano/microsized pores of the membrane after which, it is driven through a passive capillary effect over a low resistance path. When the water enters the pores, a meniscus is formed inside each pore, which is typically comprised of three characteristic regions;

(1) an absorbed region, in which a set of attractive forces exist between the liquid and the membrane wall forming the basis of the meniscus, (2) a transition region, in which the layer of water has a high curvature, forming low thermal resistance between the wall channel and the fluid, which causes better exploitation of the latent heat of evaporation and enhances evaporation, and (3) a bulk fluid/meniscus region, in which the curvature of the interface is constant, serving as a feed for the transition region in which the capillary forces are dominant [8,10,11].

Several studies investigated thin-film evaporation, with an emphasis on the interfacial transport and evaporative mechanisms; Ibrahem et al. [12] investigated the characteristics of evaporative heat transfer at a solid-liquid-vapor contact line, in which the wall temperature distribution beneath the meniscus was investigated experimentally, and demonstrated that the local heat flux at the three-phase region was 5.4–6.5 times greater than the average input heat flux of the heat source, and was attributed to a high rate of evaporation. In another study, Raj et al. [13] demonstrated the behavior of a macroscopic contact angle of dielectric fluid at the three-phase contact line, under superheated state, and experimentally demonstrated that the macroscopic contact angle increased in a micro-region where the evaporation was carried out from the meniscus. However, he found numerically that the contact angle of the liquid-vapor interface was influenced by several parameters such as evaporation, capillary, and Van der Waals forces in the evaporation region. Moreover, Xiao et al. [14] studied the effect of negative pressure using ceramic nanoporous membranes whilst utilizing alcohol as a working fluid, in which the effective fluid transport was achieved through decoupling the viscous resistance from the capillary pressure. They demonstrated that the dissipating heat flux at the liquid-vapor interface could reach 96 W cm$^{-2}$, and provided guidance for the design which shows the relationship between the high interfacial heat flux performance (1000 W cm$^{-2}$) and the membrane thicknesses. More recently, Wilke et al. [15] demonstrated the effects of different geometric parameters of anodic aluminum oxide nanoporous membranes on the heat transfer performance [11], in which the average pore diameter, porosity, and position of the meniscus in the pores of ceramic membranes were varied experimentally, and revealed that the evaporation flux enhanced when the volume porosity increased, due to the increased heat transfer area. However, they also demonstrated that the vapor resistance increases when the meniscus is pushed down within the pores. Nevertheless, it has also been demonstrated in the literature, that the thermal performance of the wicking mechanism within micro/nanosized pores were improved by utilizing a highly conductive material such as sintered copper mesh integrated with carbon nanotubes, titanium base heat pipes, and oxygen-free copper, which facilitated the capillary pressure in the pores, driving the working fluid, and achieving improved thin-film evaporation [16–18].

All of the previous studies were important to highlight pore transport mechanisms and to provide directions for high-performance membrane design. However, although important, very little has been performed in the literature to demonstrate those effects experimentally in a continuous flow system [19–22]. All of the previously reported solar-assisted evaporation experiments [23–30] involved batch-mode setups. The batch mode setups do not involve the flow of water; therefore, we suspect that the accumulation of heat in the fluid over time is likely to occur and thus the ability to obtain a real correlation between physical membrane properties and the actual evaporation flux can be influenced.

Herein, we fabricate polyvinylidene fluoride (PVDF) membranes containing variable amounts of fumed silica (FS) additive to achieve membranes for solar-assisted thin-film evaporation with different properties. Our approach had two important advantages: (1) the achievement of negligible change in thermal and solar absorbance properties across different membranes and (2) the achievement of variable pore sizes, volume porosities, and wettability. Moreover, we performed all of our membrane testing investigations in a continuous flow setup (as opposed to a batch/stagnant mode setup), which was important to eliminate heat accumulation in the bulk of the water and provide realistic correlations between the variables and the actual evaporation fluxes achievable. Thus, the mass transport properties of our fabricated membranes were the main variables in our study, and we show their direct effects on the evaporation mass flux experimentally.

## 2. Methodology

### 2.1. Materials

Polyvinylidene fluoride (PVDF, Kynar 741, Arkema Inc., Philadelphia, PA, USA) with a molecular weight of 250,000 kDa and a density of 1.78 g cm$^{-3}$, dimethylacetamide (DMAc, >99.5%, Sigma-Aldrich), and fumed silica (FS, TS-610, modified with dimethyldichlorosilane, Cabot Corp.) were used as the base polymer, solvent, and the additive, respectively. Deionized water (DI, Purite, 18.2 MΩcm) was used during the membrane fabrication and throughout all experiments. Sodium Chloride (NaCl, VWR, Leuven, Belgium) was used to create 35 g L$^{-1}$ feed water solutions, which was used in the evaporation testing experiments.

### 2.2. Fabrication of PVDF-FS Membranes

All membranes were prepared through vapor-induced phase separation, as illustrated in Figure 1. Five different membrane samples were prepared, as shown in Table 1, among which a pristine PVDF membrane without FS was tested as the reference sample.

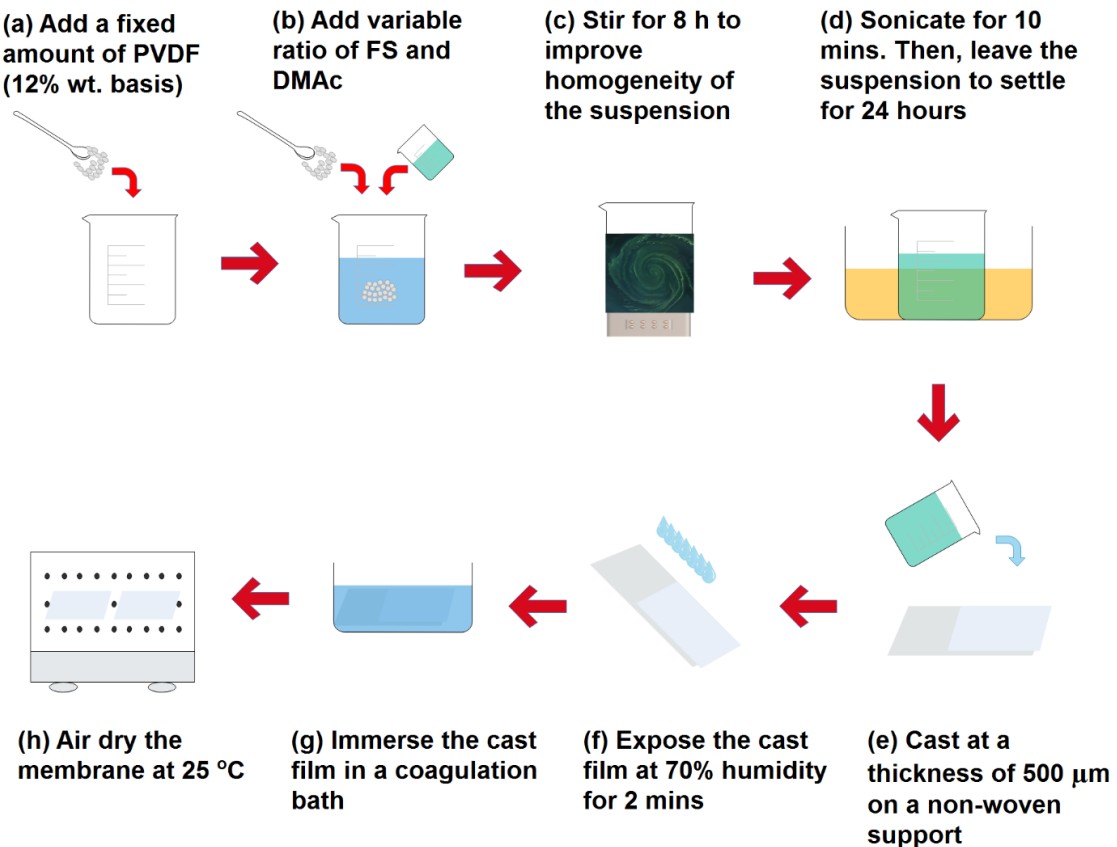

**(a) Add a fixed amount of PVDF (12% wt. basis)**

**(b) Add variable ratio of FS and DMAc**

**(c) Stir for 8 h to improve homogeneity of the suspension**

**(d) Sonicate for 10 mins. Then, leave the suspension to settle for 24 hours**

**(h) Air dry the membrane at 25 °C**

**(g) Immerse the cast film in a coagulation bath**

**(f) Expose the cast film at 70% humidity for 2 mins**

**(e) Cast at a thickness of 500 μm on a non-woven support**

**Figure 1.** Fabrication of PVDF/FS membranes; (**a**) a fixed amount of PVDF was first added into a beaker, (**b**) variable ratios of FS and DMAc were added, (**c**) stirred for 8 h, (**d**) sonicated for 10 min in a sonication bath to remove all bubbles, (**e**) membranes were cast on a non-woven support layer, (**f**) the cast layer was exposed for 2 min at 70% humidity after which it was immersed in a coagulation bath, and (**g**) the membrane was air-dried for 24 h at 25 °C.

**Table 1.** Membrane preparation parameters followed in this study[a].

| | Mass Ratios (%) | |
| --- | --- | --- |
| **Membrane** | **FS** | **DMAc** |
| PVDF | 0 | 88 |
| PVDF-FS3 | 3 | 85 |
| PVDF-FS4 | 4 | 84 |
| PVDF-FS5 | 5 | 83 |
| PVDF-FS7 | 7 | 81 |

PVDF = polyvinylidene fluoride; FS = fumed silica; DMAc = dimethylacetamide.

Dope solutions were prepared by adding a fixed amount of PVDF (12 wt %) and variable amounts of FS and DMAc into a beaker. Subsequently those beakers were stirred for 8 h at 200 rpm, sonicated in a bath-sonicator for 10 min, and then left to settle for 24 h to remove any trapped gases that may have formed during the preceding steps. All of these steps were essential to form a homogenous solution with good polymeric interlinking. Dope solutions were then cast on non-woven supports (2471, Novatexx, Freudenberg-Filter, Weinheim, Germany) using a doctor-blade (EQ-Se-KTQ-100, MTI, Richmond, CA, USA) at a wet thickness of 500 μm. The cast membrane was then exposed to 70% humid air for 2 min and then immersed in a DI water bath for phase inversion to occur. Finally, the membranes were dried at an airflow of 0.46 m s$^{-1}$ using a horizontal laminar airflow workstation (NU-340, Nuaire, Airegard, Plymouth, MA, USA).

*2.3. Characterization of PVDF-FS Membranes*

Surface morphology of the fabricated membranes was characterized by scanning electron microscopy (SEM, FEI Nova-Nano SEM-600, The Netherlands). Images were acquired under high vacuum and at a working distance of 5 mm. To avoid charging due to the polymeric nature of the samples, the membranes were initially coated with Au/Pd nanoparticles at a thickness of a 100 Å and a rate of 0.8 Å s$^{-1}$, using a Precision Etching Coating System (PECS 862, Gatan Inc., Pleasanton, CA, USA).

Volume porosity was characterized by a gravimetric method. Each membrane was wetted with a liquid of known surface tension (Galwick, Porous Materials Inc., Ithaca, NY, USA), and mass was recorded before and after the wetting procedure. These measurements were used to calculate the number of void spaces within the bulk of the membranes.

The mean flow pore size of each membrane was measured using a capillary flow porometer (CFP, Porous Materials Inc., Ithaca, NY, USA), in which dry and wet curves were generated, utilizing Galwick (Porous Materials Inc., Ithaca, NY, USA) as the wetting liquid during operation.

The wettability was characterized using the contact angle method, in which 3 μL of de-ionized water (DI) was dropped by means of a goniometer (Krüss DSA 10Mk2, Hamburg, Germany) at 24 °C.

Solar absorbance was measured by means of a spectrometer (Perkin-Elmer, UV/Vis/NIR Lambda 1050, Rodgau-Jügesheim, Germany), in which the absorbance was plotted over a range of light wavelengths (400 to 800 nm).

Thermal conductivity of all membranes was characterized using a constant thermal analyzer (TPS 2500 S) at room temperature, a thin-film sensor (7280) with a ~1-inch diameter and software modules (Slab) were utilized.

*2.4. Solar-Assisted Evaporation Experiments*

The evaporation performance of the membranes was measured using an in-house built solar-assisted evaporation setup, as shown in Figure 2. The setup consisted of a device that enables the direct contact between the flowing feed water and the bottom of the assembled membranes. The saline water feed (35 g L$^{-1}$) was circulated at a fixed flow rate of 15 mL min$^{-1}$ by a peristaltic pump

(Masterflex, Cole Palmer, Chicago, IL, USA). Other variables (humidity, temperature, and irradiation) were controlled by housing the whole setup within an enclosure. Humidity and temperature of the experimental environment were continuously measured by a multi-functional meter (EKO, MS-02). A solar simulator (Newport Corp., Irvine, CA, USA) with a xenon lamp (Ushio, Xenon UXL-16SB, XE 1600 W) was used to simulate the solar irradiation, and a total-reflection mirror was placed at a specific distance to irradiate the membrane surface with approximately 1000 W m$^{-2}$ of radiation. A reservoir containing 35 g L$^{-1}$ saline water was used as the feed and its mass was continuously recorded. Mass loss over time with and without solar radiation was recorded, and the difference between the two provided the actual evaporation mass flux through the membrane. Change in temperature ($T_{out}$–$T_{in}$) was recorded by calculating the difference between the inlet and outlet temperatures of the evaporation device over time.

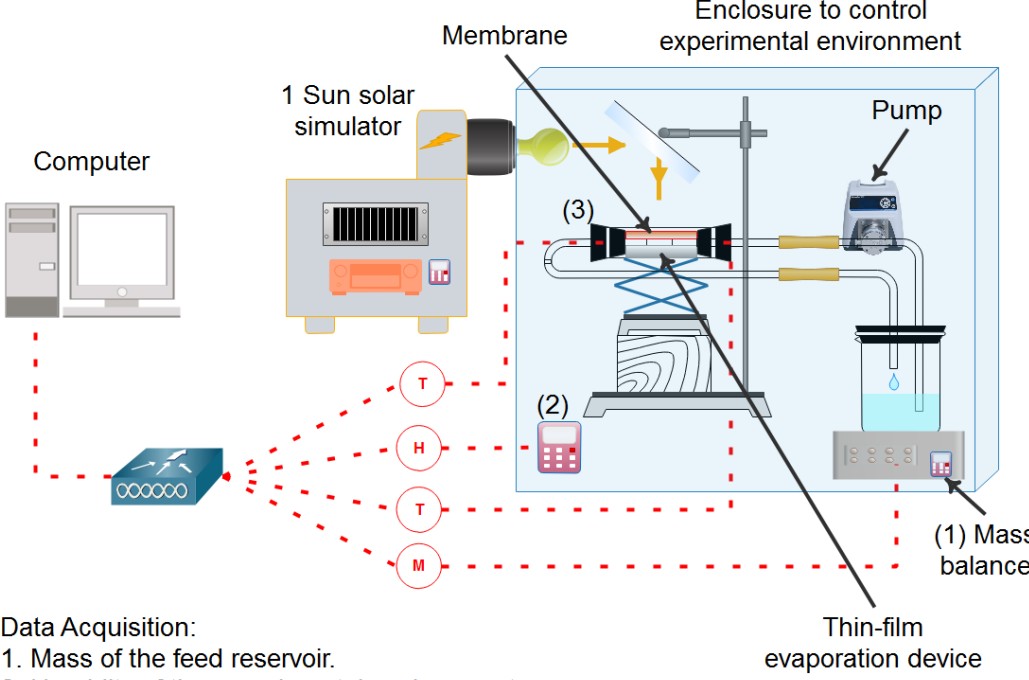

**Figure 2.** Schematic of the experimental setup illustrating the data acquisition and the connections of the device.

## 3. Results and Discussion

### 3.1. The Impact of FS Blending on the Membrane Properties

The surface morphology of the fabricated membranes was probed using SEM, and the obtained micrographs are shown in Figure 3. Cross-sectional images are shown in Figure S1 (Supporting Information). The results illustrate that all membranes were porous and that the FS particles were not homogeneously distributed on the surface of the membranes, but rather were agglomerated. The extent of agglomeration increased as the FS amount in the membrane increased, which can be attributed to the effect the FS additives have on increasing the viscosity of the casting fluid during the fabrication procedure. Moreover, the sizes of the pores were variant, suggesting that variable amounts of FS could result in variable speeds of nucleation distributed within our as-fabricated PVDF polymer layers. Furthermore, the development of the randomly distributed macropores within our membranes could be attributed to the effects of the rapid intrusion of DI within the skin layer of the PVDF membranes [1]. The images probed at lower magnifications suggest that the structure of the membranes was consistent,

and a good degree of reproducibility was achieved throughout our fabrication process. Randomly distributed dark regions across all the membrane structures suggest that the membranes follow the typical distinctive vapor-induced phase separation (VIPS) membrane characteristics, resulting in the accumulation of pores [2]. The results confirm that the addition of the FS particles had no effect on the VIPS characteristics resembled by membranes fabricated with pure PVDF.

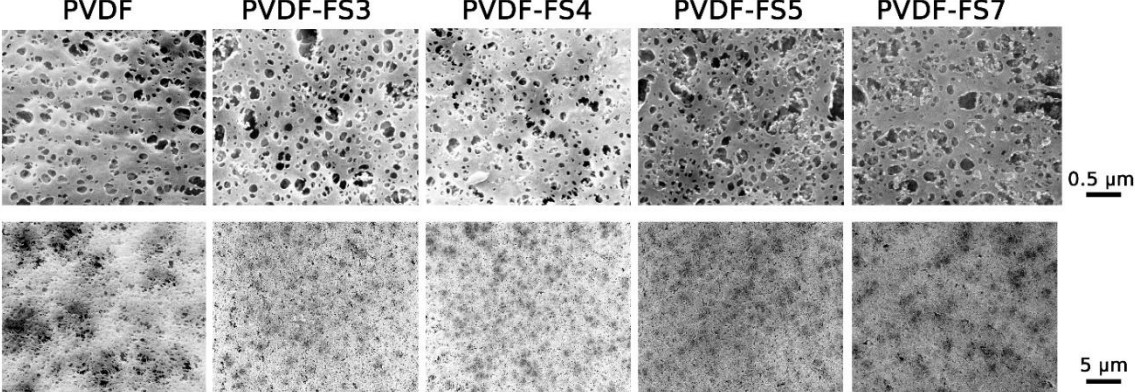

**Figure 3.** SEM images of the fabricated membranes. The results show that the FS particles agglomerated more as the amount of FS content added increased. The results also indicate that different contents of FS resulted in variable pore sizes, where the PVDF-FS4 and the PVDF-FS7 resembled the lowest and the largest pore sizes, respectively.

An excessively hydrophilic membrane can result in flooding, whilst an excessively hydrophobic membrane can interfere with the penetration of the water meniscus through the pores; therefore, a carefully designed wettability is critical to facilitate efficient mass transport from the water film into the pores of the bulk of the membrane, and thus ensure enough replenishment of water from the water film into the pores. Therefore, we performed wettability investigations, as shown in Figure 4a. The results show that the PVDF membrane (with no FS additives) resembled a contact angle of 95.5°, which decreased when small amounts of FS were added (92.5° and 92.9° for the PVDF-FS3 and PVDF-FS4 membranes, respectively), and increased when larger amounts of FS were added (109° and 113.5° for the PVDF-FS5 and the PVDF-FS7 membranes, respectively). PVDF-FS3 and the PVDF-FS4 membranes resembled similar contact angles, which were the lowest as compared to all other membranes, whilst the PVDF-FS7 resembled the highest contact angle, respectively, and could be attributed to the effect of the FS additive on the structure of the membrane. It is noteworthy here that these results indicate that FS has no effect on the membrane wettability at concentrations of 4% and below. The slight dip in contact angle seen when moving from pristine PVDF to 3 and 4% FS could be due to small changes in the membrane structure during phase inversion since these dips are less than 5% different from the pristine PVDF contact angle.

The pore sizes of fabricated membranes are of primary interest because the pores are the driveway paths in which evaporation flux occurs. Moreover, it has been reported in the literature that a higher rate of mass transfer occurs at the edges of the pores rather than their middle regions, as suggested by the perimeter effect phenomenon [3,4], suggesting that smaller pore sizes can drive a higher rate of evaporation, as shown in the literature via mathematical modeling approaches [31].

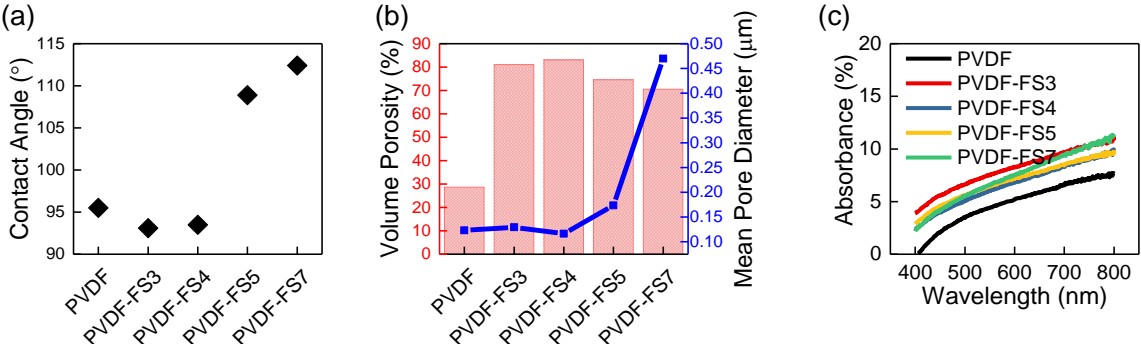

**Figure 4.** (**a**) Contact angles (**b**) Porosity and Mean flow pore diameter, and (**c**) absorbance measurements for our PVDF-FS membranes. The results show that the PVDF-FS4 membrane resembled the lowest contact angle, the smallest mean pore diameter, and the largest volume porosity, indicating that there was an optimal FS content value to achieve the desired properties. All membranes showed similar absorbance behavior over all of the wavelengths studied.

As shown in Figure 4b, out of all membranes, the PVDF-FS4 has the smallest mean pore size whilst the PVDF-FS7 has the largest mean pore size. This can be attributed to the larger interlinking effect that occurs within the skin layer of the polymer when specific amounts of FS particles are added.

Another important property of membranes utilized in solar-assisted desalination applications is their ability to absorb solar irradiation and its subsequent conversion into thermal energy through conduction and convection of the water within the membrane pores. We performed solar absorbance (Figure 4c) and thermal conductivity measurements (not shown) on our membrane sheets, and the results suggest that all membranes exhibited similar solar absorbance over 400–800 nm wavelength, all of which were slightly higher than that of the PVDF membrane. Moreover, all membranes exhibited similar thermal conductivity (0.7 W mK-1 ± 0.1), indicating that the effect of varying the FS amount on the heat transfer properties of our as-fabricated membranes is small, and providing supporting information that the performance of our as-fabricated membranes likely performs similarly in terms of heat transfer and solar absorbance.

### 3.2. Performance of the PVDF-FS Membranes in the Solar-Assisted Evaporation of Saltwater

We assembled each membrane in an in-house designed and built thin-film evaporation device and recorded the measurements (Section 2.4). As shown in Figure 5a, all membranes showed a stable flux when solar irradiation was absent, which was consistent over the entire duration of the experiment, indicating negligible fouling. When the experiments were repeated in the presence of solar irradiation (1 sun), the membranes exhibited a significant increase in flux over the first 2 h of the experiment, after which stability was achieved, confirming that the fabricated membranes were capable of utilizing solar energy to drive evaporation. Furthermore, the results (Figure 5b) show that, as compared to the PVDF, PVDF-FS3, PVDF-FS5, and PVDF-FS7 membranes, the PVDF-FS4 membrane had the highest mass flux over the entire duration of the experiment. This could be attributed to its highest hydrophilicity, facilitating more supply of water into the capillaries of the membrane, as well as its larger volume porosity and lower mean flow pore diameter, resulting in the improved capillary pumping effect (as evident from Section 3.1). Higher porosity also facilitates lower mass transfer resistance due to the larger available void volumes for water vapor transport to occur. Furthermore, we suggest that the improvement in performance was not due to any changes in vapor pressure since it was evident that all membranes exhibited similar absorption and heat transfer properties (as evident from Section 3.1). This is also evident from the negligible difference of the temperature rise across the membranes with a range of 5 to 7 °C between the lowest and highest (Figure 5c). In general, when compared to pristine PVDF, adding fumed silica nanoparticles enhanced the membrane performance and characteristics, as shown in Figures 4 and 5b. Moreover, Figure 5c confirms that the change in temperature, which is

primarily due to the accumulation of heat, is negligible. This confirms that the attained evaporation flux is directly related to the effect of the membrane pore properties, i.e., mean pore size and porosity.

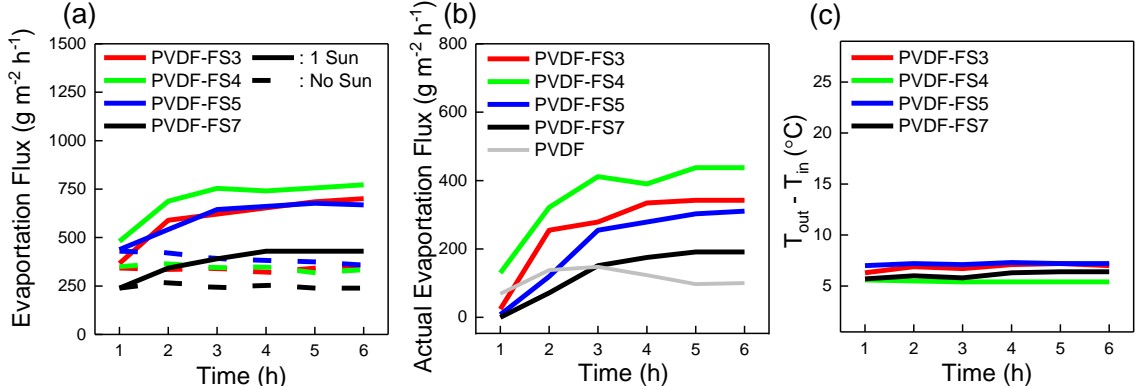

**Figure 5.** (**a**) Recorded mass flux over time with and without solar irradiation, (**b**) Calculated amount of flux driven solely due to 1 sun of solar irradiation, (**c**) Change in temperature showing that the heat accumulation due to heat transfer properties was negligible. The results indicate that all of our membranes were stable over 6 h, and that the PVDF-FS4 exhibited the highest performance as compared to the PVDF-FS3, PVDF-FS5, PVDF-FS7, and PVDF membranes.

## 4. Conclusions

We have demonstrated the improved performance of fabricated PVDF-FS membranes containing variable amounts of FS additive in solar-assisted thin-film evaporation. We showed that FS additives could be utilized to manipulate physical pore properties whilst having negligible effects on thermal and heat transfer properties. Thus, the mean pore size diameter of the fabricated membranes was the main variable in our study. Our results show that a higher volume porosity and smaller mean flow pore size increased the evaporation mass flux. This indicates that the mass transfer was governed by thin-film regions of the meniscus occurring from the small fluid velocities near the interface. We attribute the results to the increase in the capillary pumping effect through the thinner mesoporous channels.

**Supplementary Materials:** The following are available online at http://www.mdpi.com/2076-3417/9/15/3186/s1, Figure S1: Cross section images of the PVDF and FS-modified PVDF membranes. The addition of FS during the fabrication procedure disrupted the fine structure of the PVDF membranes, resulting in thicker polymeric strands, less interlinking, and a likely lower tortuosity., Figure S2: Mechanical strength of the PVDF and FS-modified PVDF membranes. The results suggest that the mechanical strength of the membranes are reasonable. As FS content increased, tensile strength and elongation values decreased. Table S1: Mechanical test results for as-fabricated PVDF and FS-modified PVDF samples. Mechanical strength properties decreased as FS content increased.

**Author Contributions:** F.A. formulated the design of the experiments. M.B. fabricated the membranes. M.B, M.A. and I.M. performed experimentation and characterization. All authors analyzed the results and contributed to the writing of the manuscript.

**Funding:** This publication is based upon work supported by the Khalifa University of Science and Technology under Award No. 8474000003.

**Acknowledgments:** This work was funded by the Cooperative Agreement between the Masdar Institute of Science and Technology (Masdar Institute), Abu Dhabi, UAE and the Massachusetts Institute of Technology (MIT), Cambridge, MA, USA.

**Conflicts of Interest:** There are no conflict to declare.

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
