# Peer review of "Effect of Pore Characteristics in Polyvinylidene Fluoride/Fumed Silica Membranes on Mass Flux in Solar-Assisted Evaporation Applications"

_applsci, doi:10.3390/app9153186_

Round 1

Reviewer 1 Report

The authors fabricated PVDF membranes incorporating Fumed Silica (FS) additive and tested its performance in the solar-assisted thin-film evaporative system. With the growing demand of new materials and technologies for clean, energy-efficient desalination technologies, the work is of great scientific importance. However, the article needs some improvement before publication. Below are minor comments:

1.     There is a formatting error in references to Tables and Figures throughout the manuscript. Please check.

2.     In Table 1: Mass of PVDF is constant so it does not need to be indicated if mentioned in methods. Also, enhance the presentation of the table by expressing the mass of FS and DMAc as the mass ratio or %wt. in a mixture with PVDF.

3.     Page 5, line 152: The concentration of saline water can be indicated before mentioning the feed flow rate.

4.     In Figure 2, the corresponding numbers for data acquisition parameters can be indicated. E.g. number 1 near the mass balance, etc.

5.     Page 6, line 172: I assume the fabrication process was reproducible rather homogenous, and this confuses with the inhomogeneity of the membranes itself due to the presence of FS. Please check.

6.     Page 7, line 187-194: If FS additive increases the contact angle, what is the reason for the initial decrease of the FS additives at low concentration (PVDF-FS3 and PVDF-FS4) compared to pristine PVDF? Clarify.

7.     Page 7, line 206-209: It seems that PVDF-FS3 has higher mean pore size compared to PVDF-FS4, which contradicts with the explanation that mean pore size increases with the content of FS. The authors might indicate the data errors to justify if the difference between the mean pore size of these two membranes is not significant. 

Please get help of native speaker to improve the language. Some examples:

1.     Page 1, line 9-10: ‘Abstract’ section, please avoid repeating the word demonstrate

2.     Also page 2, line 74-75: Avoid the repetitions ‘important’…..’although important’

Reviewer 2 Report

Please show the cross-section SEM image of each sample in Figure 4.

The reviewer strongly suggested that the author perform the pore size analysis on different region (depth) of the sample, since the topic of this manuscript is focused on the effect of pore characteristics.

In this study, DMAc was used as solvent. The amount of DMAc remained in the membrane should be specified.

From my judgement, the pore structure of FS layer may be profoundly affected by process (f) and (g). Although the authors provided the effect of the amount of FS on porosity and pore diameter, additional experiments with different humidity and exposure time and the effect on pore structure should be provided.

The contact angle of FS should be provided for comparison (by adopting the same membrane fabrication process without PVDF).

The purpose of introducing Eq. (1) is not clear. After all, the main mechanism of water transport in this study is capillary effect. The applicability of Hagen-Poiseuille equation to describe capillary effect may not be adequate.

Line 220: 3.1à3.2

Lines 238 and 240 indicate there were sections 3.1.2 and 3.1.3. However, the reviewer was not able to find corresponding sections in the manuscript. The editorial work of this manuscript should be carefully conducted.

The pore structure analysis is not sufficient. The only pore structure analysis of the proposed membrane is in Figure 4 b (porosity and mean pore diameter). Pore structure in general includes the porosity, pore size, pore size distribution, and pore morphology of a porous medium. The reviewer strongly suggested that the authors to perform additional pore characteristic analysis, and to establish the correlation (qualitative or quantitative) between the pore characteristics and water evaporation flux.

Figure 5c: The measurement method for temperature (Tin and Tout) should be specified.

The most important aspect of this study is to control the pore structure. Nevertheless, the mechanic property of the membrane is also important. The reviewer suggested that the authors to perform bending test on the membrane (e.g. tensile strength, hardness and flexibility test (ASTM F137-08)).

The durability of the membrane should be investigated.

In addition to the contact angle measurement, the reviewer suggested that the authors to perform the water uptake experiment to illustrate the water penetration effect in the membrane.

Round 2

Reviewer 2 Report

none